# Participation of young women in sexual and reproductive health decision-making in Malawi: Local realities versus global rhetoric

**Jannah M. Wigle**[1]*, **Stewart Paul**[2], **Anne-Emanuelle Birn**[1,3], **Brenda Gladstone**[1,4], **Monica Kalolo**[5], **Lumbani Banda**[2], **Paula Braitstein**[6,7]

**1** Social and Behavioural Health Sciences, Dalla Lana School of Public Health, University of Toronto, Toronto, Canada, **2** Parent and Child Health Initiative (PACHI), Lilongwe, Malawi, **3** Global Development Studies, University of Toronto, Toronto, Canada, **4** Centre for Critical Qualitative Health Research, Dalla Lana School of Public Health, University of Toronto, Toronto, Canada, **5** SOS Children's Village, Ngabu, Malawi, **6** Division of Epidemiology, Dalla Lana School of Public Health, University of Toronto, Toronto, Canada, **7** School of Public Health, College of Health Sciences, Moi University, Eldoret, Kenya

* jannah.wigle@mail.utoronto.ca

**Data Availability Statement:** The data that support the findings of this study are not publicly available due to the ethics consent form indicating that

## Abstract

Despite the global prioritization of addressing adolescent girls' and young women's sexual and reproductive health (SRH) and participatory rights, little research has examined their lived experiences in shaping their engagement in SRH decision-making processes in the global South. Further, few studies have explored how structural and societal factors influence their agency and participation. This critical and focused ethnography, informed by postcolonial feminist and difference-centred citizenship theories, conducted in Malawi (2017–2018) elicited perspectives of youth and key informants to help address these knowledge gaps. Our findings show that the effective implementation and uptake of global discourse on participation and gender equity is hindered by inadequate consideration of girls' and young women's local political, cultural and social realities. Many girls and young women demonstrate passion to participate in SRH policymaking as agents of change. However, patriarchal and gerontocratic political and social structures/institutions, and gendered and adultist norms and practices limit their active and meaningful participation in SRH decision-making. In addition, donors' roles in SRH policymaking and their prioritization of the "girl child" highlight an enduring postcolonial power over agenda-setting processes. Understanding young people's experiences of gendered participation and scrutinizing underlying systemic forces are critical steps toward realizing young women's SRH and participatory rights.

## Introduction

Global actors have increasingly championed the participation of young people, particularly young women, in decision-making processes as a critical strategy to realizing their health and wellbeing [1]. In recent decades, substantial international, regional and national commitments have been made to ensuring young people's "right to participate" in decisions that affect them, including Article 12 of the *Convention on the Rights of the Child* and Article 11 in the *African Youth Charter* [2, 3]. In Malawi, human rights-based, youth, and sexual and reproductive

participant data would not be shared outside of the research team. This study is based on extensive identifiable data from observations, interviews and focus group discussions. These data may contain information that could compromise the trust, confidentiality, and privacy of research participants, and thus will not be made publicly available. This research study was approved by the Research Ethics Board at the University of Toronto. If you have any questions or concerns or requests for data access, please contact the Research Oversight and Compliance Office – Human Research Ethics Program at ethics.review@utoronto.ca or by phone at +1 (416) 946-3273.

**Funding:** This research was supported by the International Development Research Centre under Grant #108279-010 (JW) and the Social Sciences and Humanities Research Council of Canada Grant #752-2017-1555 (JW). The funders had no role in study design, data collection and analysis, decision to publish, or preparation of the manuscript.

**Competing interests:** The authors have declared that no competing interests exist.

health (SRH) policies, and the creation of youth participatory structures at all levels of governance show progress in meeting these obligations [4–6]. Also, Malawi has committed to the global Sustainable Development Goals (SDGs), including Goal 5, which aims to achieve gender equality and empower all women and girls [7]. Despite these efforts, in 2018 Malawi ranked 172 out of 189 countries worldwide on measures of gender equality, encompassing reproductive health, empowerment (education, empowerment index and political participation), and female participation in the labour market [8]. Our study aims to explore this disconnect between the global rhetoric on gender equality and youth participation and young women's lived experiences of SRH participation in Malawi.

Gender inequity and limited representation of youth in policymaking processes are ongoing challenges worldwide [9]. The participation of girls and young women is poorly documented in youth, girlhood and human rights literature [10, 11]. "Girlhood" discourses in the global South have contested the use of "empowerment" strategies as the main solutions to what are deeply rooted, systemic inequities [12]. Scholars highlight the overlap between poverty, gender and gender-based violence in limiting girls' agency, access to SRH services and their participation in decision-making [13]. A global study with youth advocates in HIV/SRH decision-making bodies found that participation depth and quality was limited, particularly illustrating inadequate integration of voices from younger adolescents, girls and young women [14]. Moreover, to measure progress towards global objectives, such as the SDGs, empirical evidence reflecting the everyday realities of youth is lacking, particularly in the global South [15]. Notwithstanding a dearth of evidence on young women's political participation within youth, girlhood or human rights discourses [10], recent scholarship demonstrates that young women in Latin America often engage in activism and political participation, informally outside the realms of the political sphere [16]. Recent research has examined the policies, progress, and challenges to engaging youth in SRH decision-making in Malawi. Despite established policies and participatory mechanisms for youth, there is limited understanding of the underlying structural and societal factors circumscribing young people's meaningful engagement at all stages of the policymaking process [6]. This study aims to provide a deeper understanding of structural inequities and power relations encountered by young Malawian women through exploring how gender, age, and other categories of social difference intersect to influence their status in largely adult and male-dominated SRH decision-making spaces and processes.

We conducted a critical and focused ethnographic study [6], informed by postcolonial feminism [17] and difference-centred citizenship theory [18] to explore how young women's complex everyday realities of participating in SRH policymaking in Malawi are shaped by underlying structural and societal forces. This aimed to challenge current SRH policymaking practices and inform the development/implementation of gendered local, youth-driven SRH policies and programs. The terms "youth" and "young people" are used interchangeably throughout this paper. Despite recognizing diversity in young people's "gender" and "gender identity", in this study the terms gender, gender equity and gender equality are frequently employed along binary dimensions (Table 1). We also acknowledge the heterogeneity of young people's trajectories towards adulthood, including their lived realities and SRH needs. These vary substantially both between, and within, countries during the transition to adulthood, amidst significant physiological, cognitive, emotional and social changes [1, 19, 20].

## Sociohistorical context

Malawi offers a compelling context for the study of youth participation in SRH policymaking, as young people aged 10–35 years account for more than half of the country's population and they experience a substantial burden of health and SRH issues, including high rates of

**Table 1. Understanding of key concepts.**

| Concept | Definition |
|---|---|
| Adolescent, youth, and young people | Definitions of "adolescent", "youth" and "young people" vary by region and country. The term "adolescent" is frequently defined as individuals aged 10–19 years, "youth" as 15–24 years, and "young people" represents population aged 10–24 years [1]. In Malawi, youth are defined in the *National Youth Policy* as individuals aged 10–35 years; it also notes that this definition is "quite flexible" [5]. |
| Gender | Social and cultural construction of "roles, behaviours, expressions and identities of girls, women, boys, men, and gender diverse people" [21 para. 2]. It shapes individuals' self-perception, social interactions and the distribution of power and resources within society [21]. Gender identity reflects individuals' "innermost concept of self as male, female, a blend of both or neither" and may be the same or different from their assigned biological sex at birth. Sexual orientation is defined as an intrinsic and "enduring emotional, romantic or sexual attraction" to other individuals [22]. |
| Equality & equity | Equality is the provision of the same opportunities, treatment or resources to each individual or group of individuals; whereas equity reflects the differential needs and circumstances of individuals and ensures that sufficient resources/opportunities are offered to reach an equal outcome [23]. |
| Gender equity & gender equality | Gender equity is understood as the "process of being fair to women and men" and involves compensating for inequalities and structural disadvantages [24]. Gender equity leads to gender equality, or the "equal enjoyment by women and men of socially-valued goods, opportunities, resources and rewards" [24]. Although there are various overlapping/competing definitions of gender equality across its varied uses (e.g., the UN human rights to ensure "equal rights of men and women"). Throughout this article we use "gender inequality" to reflect the global development rhetoric (and tensions therein) and "gender inequity" to signal the structural oppression and discrimination of girls and young women participating in SRH decision-making processes. |
| Heteronormativity | A "set of social assumptions and norms which are based on heterosexual, cisgender experiences, influenced by social biases, privilege and stereotyping" [25] and reproduced in everyday activities and life where "gender, sexuality and heterosexuality interconnect" [26]. |
| Patriarchy | The power, control and dominance of men in social institutions and systems, and the system of male oppression and control of women [27]. |
| Power | Power represents a dynamic concept influenced by interlocking axes of domination and oppression [28]. This understanding is informed by postcolonial feminist theory and reflects the ways young people are "subject(s) of power" shaped by their legal, economic, religious, and social contexts and challenges dichotomies of "power" (e.g., "possessing power versus being powerless") [29]. |
| Policymaking & decision-making | In this study, policymaking is defined as the iterative process of prioritizing issues on the policy agenda, formulating solutions, making decisions, implementing actions and monitoring and evaluating progress [30]. Decision-making involves the adoption of a particular course of action, and represents an integral element of policymaking; it is described as the consideration of possible options, consequences and is influenced by policy aims and environment [31]. |
| Gerontocracy & Adultism | Gerontocracy refers to a political system ruled by few "old men" [32]. However, in several African contexts this understanding also integrates "traditional African respect for authority of elderly persons for their wisdom, knowledge of community affairs, and 'closeness' to the ancestors" [33]. |
| | Adultism describes beliefs that adults are more knowledgeable and superior to children/youth [15]. Children and young people are constructed as a homogenous group and an object or possession of parents/families [34]. |
| Youth participation | The active and meaningful involvement of young people in all aspects of their own, and their communities' development, including their empowerment to contribute to decisions about their personal, family, social, economic, and political development" [1]. |

unplanned pregnancy, HIV/AIDS and sexually transmitted infections (STIs), unmet need for contraception and unsafe abortion [4, 35, 36]. Further, young women are particularly vulnerable to experiencing gender-based violence, including intimate partner violence, forced or coerced sex and human trafficking [35]. In Malawi, traditional and cultural practices also shape young people's SRH knowledge, attitudes and behaviours through initiation ceremonies, high rates of child marriage and early sexual activity [37]. Moreover, an entrenched culture of adultist beliefs and gerontocratic structures in Malawi [38] has circumscribed young people's citizenship, participation and agency [18]. Research that analyzes these underlying forces, including gender inequality, and social, gender and adultist norms [20] is particularly relevant to gaining insights of the inequities impeding progress for youth SRH and participation [1].

Historical and contemporary colonial institutions have also firmly established gender as a social construct, contributed to the subordination of African women, and undermined young women's status and authentic participation [39]. African feminist Oyewuni argues that the conceptualization of "woman" is deeply-rooted in and imposed by Western discourses of "biological determinism and the linkages between the body and the 'social'" [39]. Semu [40] also argues that colonialism has undermined women's autonomy, status and roles in Malawi through the introduction of more narrowly defined gender roles and inequitable opportunities for education. Traditional Chewa women were afforded higher status as the "root of a lineage, [where] the woman was seen as a sacred vessel of life" as their power and personhood was intimately related to their ability to bear children [41]. However, several key factors starting in the 19th century are implicated in the "erosion" of this status: i) the introduction of slavery and its continued practice until the late 19th century; ii) the reinforcement of patrilineal societies under colonialism; iii) the introduction of Christianity and its pervasive influence; and iv) the introduction of a capitalist economy and displacement of women's roles in agricultural production [40]. Religion remains an important element of the research context in Malawi; most of the population is Christian (86.9%), while nearly 13% is Muslim [37]. Although Islam was first introduced to Malawi during the 16th century via traffic and trade opportunities from East Africa [42], Christianity was implicated in "crushing" the practices and tenets incompatible with its male dominance [41]. Colonial rule also transformed male-privileging norms that shaped women's autonomy and status, by limiting their educational opportunities, inheritance claims, and traditional roles in food production and agriculture [40]. In particular, the large role played by Christian missionaries under colonialism in Malawi fostered enduring class and gender educational differentials by training men as pastors, teachers, clerks and tradesmen [40].

In Malawi, both patrilineal and matrilineal structures exist amongst the 12 major ethnic groups [36], yet even within matrilineal systems women's participatory power, status and autonomy were circumscribed [40]. Despite post-independence legal reforms and human rights discourse adopted by President Kamuzu Banda which appeared to champion women and gender equity (e.g., nominating women to parliament), tangible progress for Malawian women has remained constrained [40]. These political, social, historical and economic factors have influenced the availability and type of opportunities for youth, particularly young women, to be engaged in decision-making processes [1]. However, there is limited empirical evidence on how these forces determine young people's power and lived experiences of SRH participation, in practice [43].

## Materials and methods

### Theoretical approach

A postcolonial feminist theory (PCF) and difference-centred citizenship approach (DCA) informed all stages of our research, including the development of the research questions, the

methodology and study design, participant sampling, data generation, analysis and interpretation, and knowledge translation. This combined theoretical framework exposes the "gendered, racialized, and adultist" [18] assumptions and structures that shape young people's participation in SRH policymaking in Malawi. Moreover, it emphasizes young people's agency, a "dialectical relationship between individual sense of self and collective action", as a critical driver of their participation to overcome sources of inequity and oppression [18].

PCF is a critical theoretical lens that aims to understand the "complex matrix" of historical, cultural, political and economic contexts contributing to inequities in power and health among non-Western populations [28]. Grounded in scholarship from postcolonial theorists, such as Mohanty [17, 29], and Black feminists, including Collins and Crenshaw [44, 45], scholars "at the margins" have challenged the inadequacies of, and expanded on, white/Western-centric feminist theory to (re)present all women's experiences. They also aimed to produce a novel theorization at, and of, the intersection of gender, race, class and coloniality in postcolonial contexts [46]. Postcolonial feminist scholars' integration of intersectionality conceptualizes the interlocking systems of oppression experienced by youth due to their social location (e.g., age, gender and class) and acknowledges the fluidity of these mechanisms of power under colonialism and postcolonialism [28]. Although an intersectional approach has helped us unpack and distinguish among the range of youth identities and experiences that are otherwise often homogenized, several scholars have also argued that contemporary intersectionality theory on its own is limited in its conceptualization of agency, privilege and structural power relations [47–49].

To overcome this limitation, our study utilizes a postcolonial feminist lens to understand how young people's agency and structural forces, social, political, economic, cultural and historical contexts and asymmetrical power relations have shaped young people's engagement in SRH policymaking in Malawi. A PCF approach views "culture" as a "fluid entity" to expand understandings of oppression to account for the ways in which "patriarchy, traditionalism and modernity" impact individuals' everyday realities [50, 51]. Development discourse is also critiqued by Mohanty [29] for its "reductive cross-cultural comparisons"; dominant forms of knowledge from the global North colonize young women's daily experiences and frequently homogenize the political interests of young women in the global South, silencing and subjugating them as the "Other" [17, 28]. By employing a PCF perspective, we challenge perceptions of "youth" as a universal category and young women as victims of oppressive socio-cultural structures. Young people represent "subject(s) of power" shaped by their contexts and experiences, to resist the dichotomies of "power" (e.g., "possessing power versus being powerless") [29]. This understanding frames youth as capable, active participants with agency in decision-making, in order to provide a more inclusive understanding of systemic forces of power and inequity [17]. Moreover, we understand the concept of "youth" as a dynamic and evolving category, shaped by the plurality of historical, political and social contexts and experiences [52].

Complementing PCF, Moosa-Mitha's [18] DCA offers a more inclusive understanding of young people's participation and citizenship, and challenges normative, liberal citizenship models, as these discourses and their colonial roots inadequately reflect young people's everyday social, political, cultural and historical contexts in postcolonial settings, such as Malawi [53]. "Differences" in young people's social positions (e.g. age, gender, race, sexuality and class) vary by context and represent assets of their citizenship, rather than limitations [18]; they acknowledge young people's citizenship rights and value their participation because of their identity as youth. It also recognizes youth as "differently-equal" through their informal, domestic or grassroots agency and participation, despite deviation from "adultist norms" [18]. This approach advocates for the pluralism of democratic values (e.g., rationality, individuality, and universality) to generate a framework articulating the "multiplicity of forms of subordination"

within social and power relations due to gender, race, class, sexuality, and other power relations [54]. It also recognizes young people's agency, expressed as collective action and participation against oppression, as a key strategy to create space and shape their lived realities [18].

Conceptualization of typologies or frameworks also has dominated literature on youth participation, with relatively limited emphasis or empirical evidence of young people's everyday realities of SRH participation [55–59]. Considering the dearth of literature of the structural and societal determinants that shape young people's opportunities for, and experiences of, participating in SRH decision-making [43], this research–informed by postcolonial feminist and difference-centred citizenship theories–employed a focused critical ethnographic approach to generate empirical insights of youth participation in SRH policymaking in Malawi. It represents a significant and timely research contribution, as it is the first study to explore young people's lived experiences of participating in SRH policymaking, not only in Malawi but also, to our knowledge, within sub-Saharan Africa.

## Methodology

A critical ethnographic approach was used to generate data about the lived experiences of young people participating in SRH decision-making processes. Informed by postcolonial feminist and difference-centred citizenship theories and reflexive practice, critical ethnography aims to expose underlying social structures, power relations and their impact on human agency [60]. Critical ethnography evolved as a methodological approach in response to critiques of "traditional" ethnography for its 'Othering' and colonizing approach to knowledge generation, lack of researcher reflexivity and its inadequate consideration of power relations within social and cultural structures and groups [61]. Further, our approach was "focused" in the sense that a focused ethnography seeks to answer specific research questions in applied social research within "fragmented and specialized" areas of inquiry in order to challenge normative assumptions about youth SRH participation. It involved generating data during intensive and defined periods of fieldwork to explore our specific research interests [62]. Our focus on understanding young people's roles within SRH decision-making illustrates that culture is ubiquitous and "unbounded" [60]. In particular, our combined critical and focused ethnographic approach aimed to explore how the culture of youth participants' everyday realities of SRH decision-making are shaped by broad structural and societal forces, including class, patriarchy and racism, in order to address power inequities and advance social justice [50, 60].

Fieldwork was conducted in Malawi from October 2017 to May 2018, in Nkhata Bay (north), Dowa (central) and Zomba districts (south) and the capital, Lilongwe. Over 85% of the population in Malawi live in rural areas [36], where Dowa and Nkhata Bay represent largely rural regions, and both Zomba and Lilongwe are composed of rural and urban settings. We were hosted by a non-governmental organization (NGO), the Parent and Child Health Initiative (PACHI). Ethics approval was obtained from the University of Toronto Health Sciences Research Ethics Board in August 2017 and the National Health Sciences Research Committee in Malawi in October 2017. Local written approval to conduct research in Dowa, Nkhata Bay and Zomba districts also was obtained. An ethics deviation and amendment was submitted to the University of Toronto and approved in November 2017 to increase the size of focus groups from 4–5 individuals to 5–10 per group due to interest by participants. Informed, written consent (in English or Chichewa) was obtained from all study participants.

## Reflexivity

Reflexivity about our "creative presence" [63] as researchers conducting a critical ethnography in Malawi underscores the importance of exploring the "complex (and sometimes

contradictory) perspectives on privilege and difference" [64]. Our positionality reflects how differences produced along multiple dimensions, such as age, gender, ethnicity, nationality (or geographic provenance) and class, inherently shaped the power dynamics with research participants, their representation and knowledge generation [65]. For example, JW's social identity as a Canadian middle-class adult white female doctoral student and her previous work experience on a maternal and newborn health project in Malawi influenced her motivations, interests, assumptions and relationships to both the substantive and geographic areas of focus for this research. We also grappled with our positionality and the decolonial imperatives facing researchers from the global North working in in the global South. Therefore, we aimed to conduct ethical and deeply reflexive research which promotes and (re)presents young people's lived experiences of SRH participation. Lived experiences relate to young people's unique representation and understanding of experiences, choices and options, and their perception of knowledge; it is shaped by diverse factors including race, class, gender, sexuality, religion, and other elements that impact their everyday realities [66].

This study was conducted by a team of Canadian (JW/AEB/BG), Canadian/Kenyan (PB) and Malawian researchers (SP/MK/LB), in order to produce representative knowledge that extends beyond our respective cultural borders. This research was conducted as part of JW's doctoral research; interviews with key informants were led by JW and supported by SP, a male Malawian youth (under 35 years) for interviews with adult community and traditional leaders. JW also had established, long-term relationships with colleagues at the host institution PACHI, which helped facilitate access to SRH policy spaces, meetings with key informants, and informal knowledge exchange. Data generation with youth was conducted by JW, SP, and a youth researcher (including MK). Other members of the research team (AEB/BG/PB/LB) provided guidance, oversight and feedback on research design, analysis, and manuscripts. Although English is the official language in Malawi, Chichewa and Tumbuka are spoken widely. Participants used their preferred language; most FGDs with youth were conducted in Chichewa (one was primarily in English), and half of the youth were interviewed in English. Sharing details of our social identity/location with participants, describing the study, obtaining participants' consent and framing youth as experts on their lived experiences were elements of building trust and rapport between researchers and participants. Anonymity of responses in interviews was assured, and FGD participants were asked to maintain confidentiality and respect each other's privacy. We also engaged six youth researchers (aged 16–24 years; one male and one female per district), with experience in youth- and SRH-related activities in their communities to support fieldwork activities, establish trust with youth participants, navigate power inequities between adult researchers/youth, and co-construct knowledge across dimensions of age and gender. Youth researchers were recruited via the host institution's website, social media (e.g., WhatsApp) and local networks, as well as the district youth offices in each region. Candidates were interviewed and selected, and provided basic training in qualitative research methods, including co-moderating FGDs, probing and notetaking. Youth researchers were also involved in disseminating the recruitment poster with their peers and networks, and engaging in peer debriefing sessions after each FGD.

However, we recognize that youth researchers' engagement in research stages prior to data generations (e.g., research focus, objectives and tools) and their sustained involvement post-fieldwork in analysis and knowledge translation (e.g., coding, conceptualization and theorization) was limited. One youth researcher (MK) and research assistant (SP) remain engaged in writing and producing manuscripts for publication. Power relations between JW and the Malawian research team were also critically considered, as SP acted as an interpret, translator and "cultural guide" to understanding local culture and context, and is/was an active participant throughout the research process [65].

We also critically reflect on the language employed throughout our work. For example, the terms "stigma" and "taboo" represent popular, often naturalized terms in SRH literature. Despite substantial variation in how "stigma" is defined and applied to substantive areas of inquiry, it is also critiqued for its individualistic focus, limited consideration of structural influences and connection of social labels to "undesirable" attributes to separate "us" from "them" [67]. The use of "taboo" is also inherently problematic, often defined by society and social norms to describe and control the "permitted and the prohibited, the do's and don'ts" [68]. Although, in this study, both youth and key informant participants frequently employed these terms, we have chosen to use these terms with quotations to reflect these challenges and to critically interrogate their meaning.

## Participant sampling and recruitment

Purposive sampling strategies employed maximum variation and "snowball" sampling [69, 70]. Sampling of youth aimed to capture diverse social positions (e.g., age, gender, education and geographic location), and levels of participation (e.g., community/district/national/international). We recruited 46 youth to engage in FGDs, and 30 in semi-structured interviews with an open-ended drawing exercise. Engaged youth self-identified as either currently participating in SRH policy spaces from community to international levels, or not currently participating in SRH policymaking. The latter group includes out-of-school youth or youth experiencing homelessness [6]. Youth participants were on average 21 and 22 years for FGDs/interviews, respectively. Most had completed some secondary education, and were unmarried without children (Table 2). Recruitment posters in both English/Chichewa were shared in print (e.g., at local youth centres, district youth offices, local/international NGO/CSOs working in youth/SRH/participation and university student unions), and electronically through email and social media. Youth researchers also distributed posters through their networks,

**Table 2. Demographics of youth participants.**

| Characteristic | Focus Group Discussions | Interviews |
|---|---|---|
| **Mean Age** | Both sexes: 21 | Both sexes: 22 |
| | Female: 20 | Female: 21 |
| | Male: 22 | Male: 22 |
| | Range: 16–24 years | Range: 18–24 years |
| **Gender (N)** | Females: 26 | Females: 15 |
| | Males: 20 | Males: 15 |
| **District** | Dowa: 8 females, 5 males | Dowa: 5 females, 5 males |
| | Nkhata Bay: 8 females, 8 males | Nkhata Bay: 5 females, 5 males |
| | Zomba: 10 females, 7 males | Zomba: 5 females, 5 males |
| **Level of Education Achieved** | Lowest: Form 2 (two years of post-secondary education) | Lowest: Form 2 (2 years of post-secondary education) |
| | Highest: currently studying undergraduate degree (4th year) | Highest: currently studying undergraduate degree (3rd year) |
| **Marital Status** | Unmarried: 33 | Unmarried: 29 |
| | Married: 0 | Married: 1 |
| | Unknown: 13* | |
| **Youth with Children** | No children: 40 | No children: 25 |
| | 1 child: 5 | 1 child: 4 |
| | 2 children: 1 | 2 children: 1 |
| **Total** | 46 | 30 |

* The demographic form used in Dowa district did not ask this question

including youth clubs, schools and communities. Participants were compensated for their time commitment and transportation, and provided a drink/snack.

We also sampled key informants, including: community and traditional leaders (e.g., religious leaders, traditional chiefs or traditional/sub-traditional authority leaders), district/national policymakers, NGO or civil society organization (CSO) representatives and multilateral/bilateral donors. Key informants provided a comparative context for data generation/interpretation of young people's lived realities of SRH involvement. Thirty-two semi-structured interviews were conducted. Several key informants also identified as "youth", and were described as "youth key informant", with data analyzed alongside adult decision-makers.

## Data generation methods

Multiple research methods were employed to generate data, outlined in Table 3. Together these approaches informed the conceptualization and theorization of our results.

## Data analysis

The analytic strategies we employed included: reflexive, thematic analysis [73], and document and visual analyses. We engaged in the six stages of reflexive, thematic analysis outlined by Braun and Clarke [73], as we: i) reviewed the data; ii) generated initial codes; iii) generated

**Table 3. Data generation strategies.**

| Research Method | Description |
|---|---|
| Moderate participant observation (n = 11 meetings) | • Observed community, district, and national meetings.<br>• Our presence as researchers was known by attendees, but we did not actively participate in policy proceedings [71]. |
| Document analysis (n = 8 policies) | • Policies/strategies were identified by searching governmental/NGO websites and publications.<br>• Items selected for analysis based on the focus on youth participation/SRH, most recent version/publication, and availability in English. |
| Focus group discussions (FGDs) (n = 6, with 46 youth participants: 20 male/26 female) | • Conducted FGDs across three districts<br>• One FGD was conducted for each males/females (youths aged 16–24 years)<br>• FGDs aimed to provide an understanding of perspectives on key concepts (e.g., participation, human rights and citizenship), underlying issues of participation (e.g., gender and class), and to anchor our work empirically.<br>• Research tools were informed by our PCF/DCA and used to explore concepts related to young people's social identity, micro (individual) power, experiences of SRH participation and citizenship, and their belonging and membership. |
| Semi-structured interviews with open-ended drawing exercise (n = 30 youth, 15 male and15 female) | • Conducted across three districts/regions (youth aged 16–24 years).<br>• Drawing represented a complementary research strategy to explore young people's lived experiences of SRH participation and produce/share knowledge with a visual outcome [72]. Youth were asked to visualize what it is like to participate in SRH policymaking, including how they felt about participating, who they worked with or where. |
| Semi-structured interviews with key informants (n = 32) | • Conducted with community leaders (e.g., religious leader, teacher and traditional chief) (n = 5), district and national policymakers (n = 10), representatives NGO/CSOs (n = 10), and bilateral/multilateral donors (n = 7). |
| Reflexive fieldwork journal | • Documented emerging ideas/biases and provided an analytic trail. |

initial themes; iv) reviewed themes; v) defined/named themes; and vi) produced written inter-pretations. An initial code list was developed based on our theoretical framework, literature, and in vivo codes (e.g., key words/processes/events) and used to code data (e.g., FGDs, observations and fieldwork journal) [74]. JW coded all interview/FGD transcripts, observation field-notes and fieldwork journal to "organize, manage and retrieve meaningful bits of data" (Coffey & Atkinson, 1996, p. 26). All interviews/FGDs were audio-recorded with participants' permission and transcribed, and Chichewa data was translated to English. NVivo 12 was employed to manage and support data analysis. Analytic summaries/memos documented coding; codes were developed into higher order categories and concepts, and used to generate themes in our work. Findings from FGDs and interviews are presented together due to the overlapping discussions and themes generated across these methods. Conceptual mapping was used to explore relationships and identify overarching themes within our data [75].

Analysis of SRH- and youth-focused policies and strategies was guided by our adaptation of the WHO *Gender Assessment Tool* [76], to evaluate the level of "youth responsiveness" (e.g., definition, framing and prioritization of youth) [6]. We also employed a critical visual method-ology to analyze drawings and reflect upon the participant, researcher and drawing. Drawings were analyzed alongside participants' descriptions to explore meanings produced, to reflect on the image and significance to their drawing and statements [72]. Mitchell and Sommer [77] underscore the "power of the visual to represent what is not easily put into words", and how youth articulate their own experience of relative power within the study.

## Results

Our analysis theorizes how patriarchal and gerontocratic forces and structures in Malawi converge to shape the lived experiences of young women's participation in SRH policymaking. Themes include: i) gender equality rhetoric; ii) donor power and discourse; iii) patriarchal culture and norms; iv) "taboo" and controversial nature of SRH; and v) young women as agents of change.

### Gender equality rhetoric

Participants acknowledged significant global rhetoric and national commitments towards addressing gender equality, yet underlying patriarchal attitudes within society and SRH decision-making processes continue to hamper bona fide participation of young women. The disconnect between unrealistic global/local expectations of young women's participation and the underlying systemic forces was summarized by a national policymaker:

> ". . .People expect a girl to act miraculously differently from where she's coming from. . .the commitment is there to create opportunities, but. . .we overlook the fact that the foundations of our society are unequal. And nothing above it could be expected to be equal if this not addressed." (Key Informant, National Policymaker)

Despite mainstreaming or integrating gender in policies and public actions designed to achieve gender equity in the workplace/polis, significant disparities in female participation were highlighted by both youth participants and key informants. A public campaign for the *Gender Equality Act* (GEA) in Lilongwe paradoxically illustrated inequity by advocating for 40/60 representation, with women lacking significant leadership positions within that proportion (Fig 1). Although at the time of research the GEA had been in place only four years, many key informants already flagged concerns around its implementation and accountability. They pointed to organizations and political bodies claiming to achieve gender "parity" by recruiting

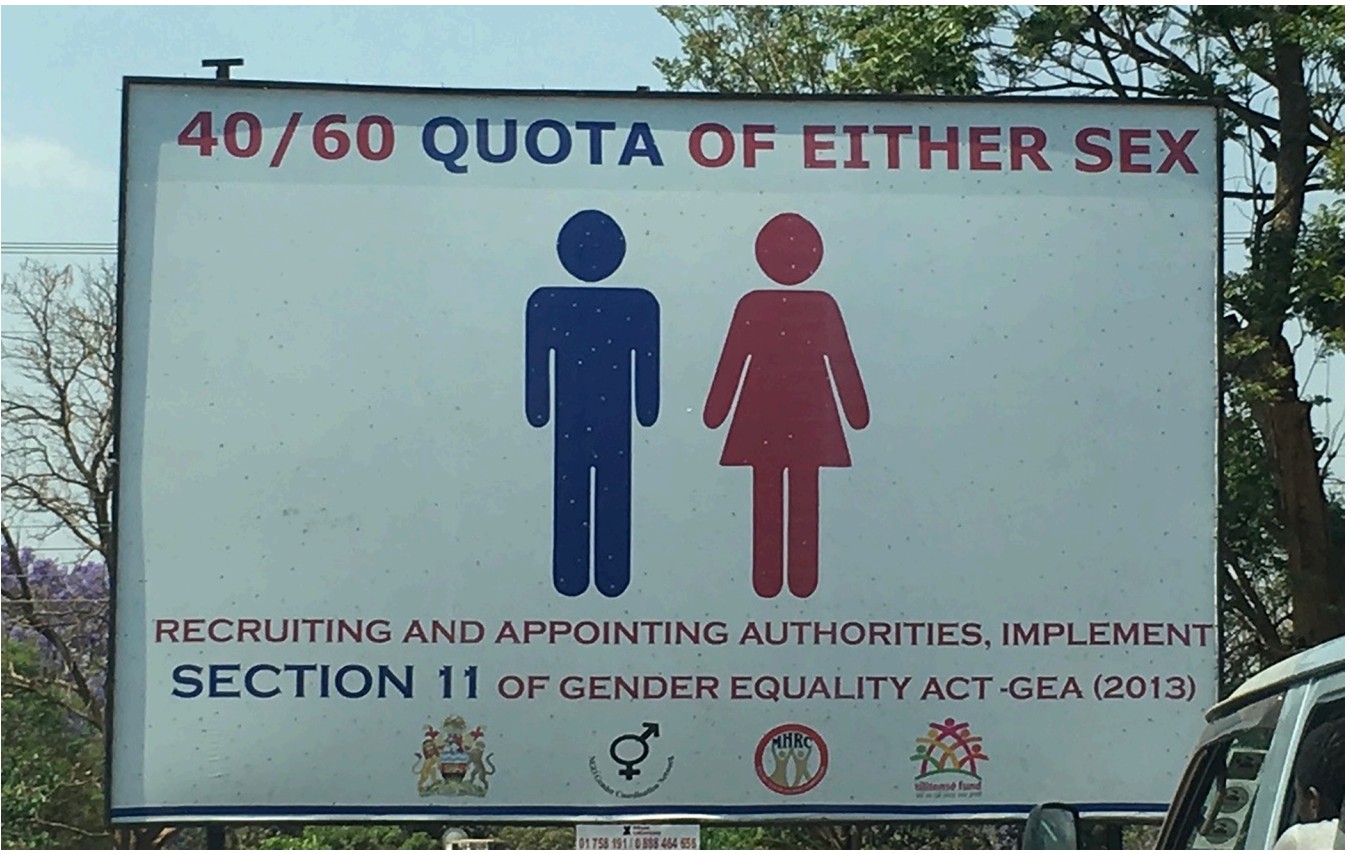

**Fig 1. Public campaign for gender equality in Malawi.**

women to administrative or entry-level positions but failing to ensure equitable leadership opportunities for women. Furthermore, few national youth, SRH or health policies offer data on youth participation or SRH disaggregated by sex, nor acknowledge the systemic exclusion of young women in decision-making spaces.

Almost all youth participants and key informants emphasized the importance of equal participation of both male and female youth in SRH decision-making spaces and processes. However, in practice young women experienced gendered and inequitable participation and leadership at all levels of governance shaped by broad gerontocratic structures. The involvement of young women occurs amidst limited representation of adult, female policymakers in national and subnational governance spaces, according to both youth participants and key informants. Although women frequently championed the implementation of SRH policy at district/national levels (e.g., female program officers in the national Reproductive Health Directorate or district Safe Motherhood Coordinators), district Youth-Friendly Health Services (YFHS) coordinators were often men. A village headwoman described the gender imbalance and the effects of this gendered assignment of health domains:

> "*Young girls are not taking part more because of what's happening at the higher level. So, it's like men are restrictive of giving opportunities for women, and that is kind of trickling down to the younger ages.*" (Key Informant, Village Headwoman)

Some efforts to ensure gender parity within national youth participatory structures (e.g., national youth parliament) were reported, yet informants identified poor accountability of

female representation in SRH policy spaces. At the district level, only one of 28 district councils had a female youth representative. Local youth participatory structures were also dominated by older male youths, as they often had established youth clubs/organizations and refused to relinquish their leadership roles. This was referred to as *"founder syndrome"* by a multilateral donor. These gendered and age-based differences in leadership further jeopardized the accessibility and appropriateness of young women's engagement on SRH issues. Although several youth participants indicated that male and female youth participate equally, they simultaneously cited inequities and used oppressive language to describe young women's SRH involvement. For example, one male youth shared that "*girls too need to be able to lead the group, for instance, should all male leaders fall sick."* Similarly, a female youth described local examples of gerontocratic structures, attitudes and discrimination:

> "*Chiefs say 'a girl cannot say this. A girl cannot do this. A girl cannot be a leader. A girl cannot do that'. So, if I try to talk to them they still say, 'you're a woman, [Name], you cannot be telling us what to do."* (Female Youth Interview Participant)

Young women's participation in SRH decision-making processes is intimately related to and dependent on gender equity and fulfilling their SRH rights.

## Donor influence on youth and SRH policy

Donors play a key role in shaping gendered SRH discourse and policymaking in Malawi, as well as young women's SRH and participation. Their priorities, political ideologies, money and power enable them to wield significant influence over local policies targeting SRH among young women. Despite evidence of national political will for advancing gender equality and youth participation, the donor-driven nature creates a power asymmetry in SRH decision-making which inadequately reflects the local context, priorities and norms, and undermines meaningful representation of young women in communities. A female youth key informant described the incongruency between donor priorities and programs, and local youth SRH needs:

> "*If a donor comes to Malawi, there are specific things that they would want to achieve, and you can't just bend that because you want young people to be fully involved. So, they also have their own agendas, which sometimes, I don't think it's in the interest of every young person in the country."* (Youth Key Informant, CSO)

For example, several key informants identified the introduction of the *Adolescent Girls and Young Women Strategy* [78], and submission of the *Termination of Pregnancy Bill* to Parliament as recent examples of SRH policymaking, championed largely by bilateral donors (the United States and the United Kingdom, respectively). These donor-driven SRH policy efforts were not fully owned/endorsed by the Malawian Government and involved limited consultation of young women.

Beyond donors' engagement in SRH policymaking, their female-focused development discourse has increased programs and initiatives claiming to improve the health and wellbeing of the "girl child" in Malawi. In our research, several male youth participants expressed jealousy of increased opportunities for young women to pursue education and engage in SRH decision-making. Despite this articulated favouritism, few female participants believed this meant their opinions were sufficiently respected or valued. Moreover, they reported insufficient emphasis on addressing inequitable power relations, including targeting young men as the

"perpetrators" to disrupt systems of patriarchal oppression and discrimination. A youth key informant shared concerns regarding this development approach:

> "*There's a lot of girls' interventions and how about the 'perpetrators'?. . .If [males] are not involved, if they are not given that information, it's more or less like one person is equipped with the information and somebody else has nothing.*" (Female Youth Key Informant, CSO)

## Patriarchal culture and norms

Girls' and young women's low status and limited power circumscribed both opportunities for their engagement, and respect for their contributions. Both youth participants and key informants reported bias, as "*males are preferred, like they're wiser than girls*", according to a female youth participant. The exclusion of young women reflects an intersection of age and gender discrimination and was voiced as a concern by a female youth:

> "*It's because of the culture. The people that are in the government, they come from different backgrounds of culture. They don't trust youth. They don't trust women.*" (Female Youth Interview Participant)

Both male and female participants reported that families, community leaders, healthcare providers and decision-makers often limit young women's access to SRH information, services or participatory spaces. Youth participants and key informants of both genders described older, male decision-makers as the "*gatekeepers*" or "*cultural custodians*" of societal (e.g., political and religious) institutions and SRH policy spaces, illustrating systems of gerontocracy discriminating young women due to their age and gender. Gender-based restrictions also required gatekeepers' permission or support to participate due to safety concerns (e.g., travelling or walking alone to youth club meetings) also limited young women's freedom and mobility. A youth key informant described elders' refusal to recognize young women's autonomy and capacity to engage in SRH decision-making; this sidelined young women based on their intersecting age and gender assumptions and chauvinism:

> "*They are all elderly people that will not create space for young girls to articulate their issues. And most of the time when an activist [stands] and speak[s] about sexual reproduction in the rural areas they would say, 'this person. . .this young girl is out of sense', because. . .she's talking about things that we are not supposed to be talked in the open, right?*" (Youth Key Informant, CSO)

Gender norms and societal expectations shape the status of women from girlhood through adulthood, and the roles available to them in their families and communities at each stage of life. The gap between ensuring young women's active and meaningful participation in SRH decision-making and Malawian women's traditionally prescribed roles are outlined by young people:

> "*. . .There is still a large dominance of [men's] 'traditional' beliefs, so in as much as they are theoretically in support of female participation and female empowerment there's a lot of traditional male roles that play a huge part in influencing male and female behaviour. . .so there's still a lot that we need to do.*" (Female Youth Interview Participant)

Young women are expected to occupy "private" spaces in society, and must balance their participation with competing responsibilities, including domestic labour, subsistence farming

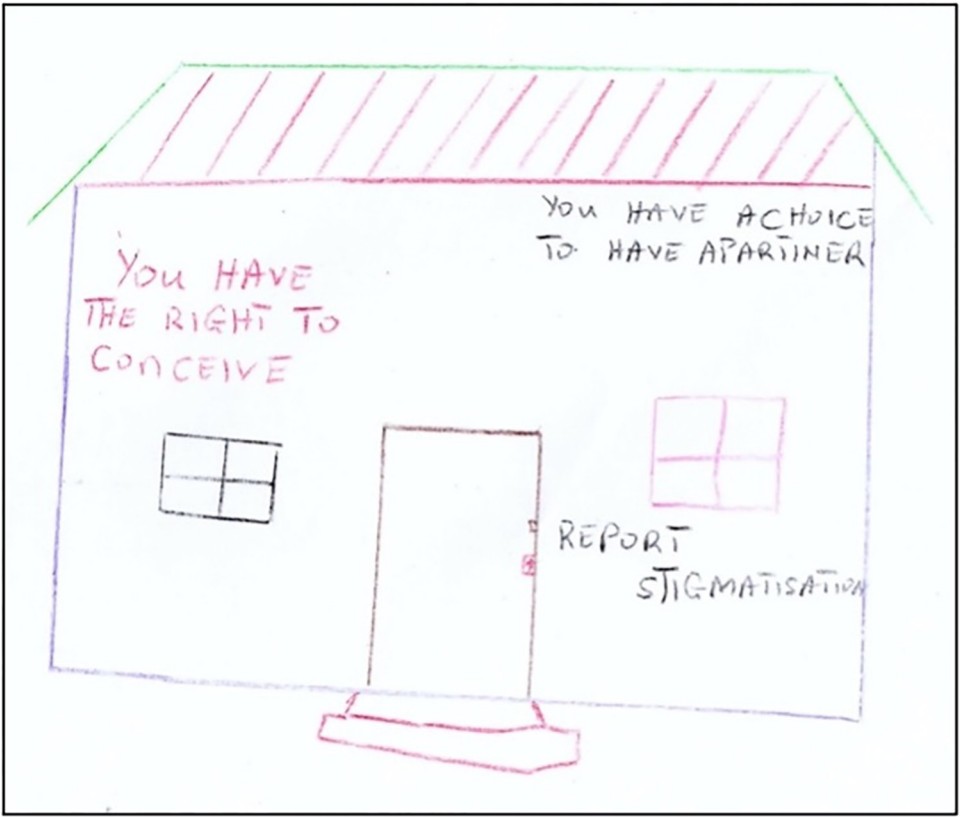

**Fig 2. Open-ended drawing: *"You have the right to conceive"*.**

and childcare. Key informants indicated that gendered parenting practices shape young people's trajectories and participation; boys are groomed to engage in decision-making, while girls are primed for domestic duties and considered *"second-class"* citizens. Several participants also outlined young women's role in maintaining local culture, through marriage/childbearing. According to a bilateral donor, young women's *"ability to bear children and be an active member of your community"* linked their responsibilities for social reproduction to their citizenship, power and participation. Social and gender norms reinforce the interpretation of young women's SRH rights, visualized by a male youth participant (Fig 2). They described young women's *"right to conceive"* involving pictures/slogans drawn on local buildings depicting having children and getting married. These are a typical form of promotions in many Malawian communities; however, these gendered and restrictive slogans may be interpreted as refashioning and naturalizing young women's "human rights" with expectations of their social reproduction.

Throughout our study, many male participants described young women as "shy" and lacking the confidence to discuss SRH issues. For example, a male youth stated:

> *"Women are so shy to discuss some issues, like they can't discuss issues like using condoms. They can't say that, so they are so shy. That's why most of the groups are dominated by men."* (Male Youth FGD Participant)

This labelling silences young women, as their political invisibility and (in)ability to participate is attributed to individual behaviours, rather than systemic patriarchal norms and attitudes. Both youth participants and key informants revealed that young women, particularly in

rural communities, are encouraged to censor their participation and perspectives (depicted as shyness by those in power) due to gender norms and the "stigmatized" nature of SRH. However, young men were often able to openly discuss issues of SRH without discrimination.

Various cultural practices were identified as harmful by youth participants and key informants, including initiation ceremonies ("*kusasa fumbi*") or arranged marriages for pre-pubescent girls to an adult male in the community ("*kutomera*"). Key informants emphasized men represent the custodians of these traditional practices, with the power to stop them. They viewed these practices as forms of gender-based violence, subjugating young women and restricting their SRH participation. Although key informants cited a decline in prevalence of these customs (Rates of child marriages and early pregnancies have increased during the COVID-19 pandemic due to school closures and public health restrictions [100].), many are still widely practiced and represent formative experiences that shape young people's SRH knowledge/behaviours. Few young women shared personal encounters with these practices, yet they were raised frequently by participants in interviews/FGDs. Nearly all youth interpreted cultural practices, particularly child marriage, as detrimental to girls' SRH and participatory rights. Young people relayed examples of their own agency to prevent child marriage in their communities, as a male youth indicated *"if a girl gets married aged 17 or below, we go as a youth club to get her out of that marriage."* Young people's efforts were recognized by multiple key informants, with one district policymaker celebrating *"the youth have really fought so hard to make sure that nobody gets married early."*

## Shame, "taboo" and "stigma"

Many "conservative" or "traditional" values relating to sex and sexuality shape Malawians' perspectives on the appropriateness of young people's, particularly girls', engagement in SRH policymaking. The SRH and health issues young women face (e.g., menstrual hygiene, contraception, and access to safe abortion services) remained "stigmatized" and inappropriate topics of discussion. Youth participants expressed unease in speaking about SRH issues with their parents, and similarly many key informants were not comfortable speaking with their own children, despite their experience working in the youth SRH arena. These perspectives and the stigmatization of SRH issues limited young women's ability to access health services and be engaged in SRH decision-making, as illustrated by a youth FGD participant:

> *"I feel like we live in a country whereby sex and sexuality is highly taboo, so it's not really a subject that young people—even if they will be given the chance—there would be FEW brave young people that would actually step up in parliament to give a presentation on sex and sexuality, that I can assure you [sounds of agreement from other FGD participants]." (Female Youth FGD Participant)*

Judgment and negative attitudes towards young women participating in SRH policymaking, due to their age and gender, substantially limits their ability and motivation to be involved. Female youth participants reported being shamed or judged for speaking out on SRH-related issues with community leaders. Young women who defied cultural and gender norms by engaging in SRH policymaking (e.g., attending local youth clubs or being an SRH peer educator), are often labelled *"prostitutes"* or *"whores"* by both adults and peers. Both male and female youth participants extended this discrimination to young women accessing SRH services or using contraception, whereby they are considered *"promiscuous"* and subjected to negative stereotyping. Many female participants' motivation for involvement was questioned by peers and elders, who negatively portrayed or shamed them for even associating with other girls engaged

in SRH decision-making. This highlighted a double standard, as young men did not encounter similar scrutiny. One female youth described her fear of judgment:

> "*I'd feel ashamed. There are elders during such meetings, perhaps elderly men and the chief could be there—for you to speak about such things the chief might think, is she not ashamed to say that. . .Where did she learn about this? [Laughs] The chief might also think perhaps she is a prostitute?*" (Female Youth FGD Participant)

Several key informants indicated that "*in the wake of the HIV pandemic*" (Community Leader) HIV/AIDS and SRH remain critical public health priorities, which has helped to decrease the shame or discomfort associated with discussing these issues.

## Young women as agents of change: Challenges and prospects

Young people's participatory and SRH rights are interdependent, as fulfilling one is critical to the realization of the other. Youth participants outlined that access to quality YFHS and meeting their SRH rights was essential to their active and meaningful participation in decision-making. Child marriage (under 18 years) and adolescent pregnancy were identified by youth and key informants as barriers to young people's wellbeing, and limited their SRH involvement. According to youth participants and key informants, young women who are married and/or have children are no longer considered "youth" and their continued engagement is inappropriate; this standard did not apply to young men. This highlights the disconnect between the definition of "youth" in the *National Youth Policy*, as compared to young women's lived, contextual realities of participation.

Female youth participants indicated that knowledge of their SRH rights and the availability of accessible and appropriate SRH services were essential to their roles as peer mentors. They underscored the importance of taking ownership of their bodies and affecting change to realize their SRH and participatory rights:

> "*I mean, there will be [a] change. People will be able to know about their rights and their own bodies. They will be able to decide. They will be able to do whatever they want to do.*" (Female Youth Interview Participant)

Many youth and key informants described young women's access to SRH information and education as an important determinant of their involvement in SRH decision-making, as well as their future economic trajectories. A village headwoman, a local traditional leader, identified that inequitable educational attainment is "*what makes a woman not able to stand in a group where men are also present.*" Young women's ability to access quality education is inherently shaped by their socioeconomic status. While youth and key informants hailed the introduction of free primary education in the 1990s, many youth participants identified secondary school fees as a major barrier. The establishment of the National Girls Education Strategy, which allows young women to return to school after pregnancy/childbirth and national legislation to raise the legal age of marriage to 18 years have helped support young women's educational status, yet their implementation remains a challenge. Female youth participants indicated that education was critical to understanding their SRH and participatory rights and overcoming structural barriers to their involvement in decision-making processes.

> "*I'm able to read. I'm able to write. I'm able to understand. I'm able to judge things. . .My education has opened me up to a number of opportunities. I'm able to comment on issues that affect me.*" (Female Youth Interview Participant)

Experiences of poverty and unemployment challenged all young people's participation in SRH decision-making, as they prioritized basic needs before engagement. Economic status also influenced the accessibility of SRH services, including abortion services, as a national policymaker outlined that *"rich people always find a way of getting their way"*. Engaging in economic activities, such as entrepreneurship, afforded young women the confidence, status and independence to challenge gendered norms and participate in decisions that affect their SRH. A female youth described this relationship:

> *"I have a fishpond, I have beehives. When we sell those things, we have our own [bank] account. . .we have an income, so that we don't lag, so that I don't beg from my parents. Because if I am going to beg from my parents, it's going to be like, 'you have to get married'. I don't want that. . .I still want to work more on this, like [with] lots of youth about their sexual rights. I try to make young people independent as well." (Female Youth Interview Participant)*

Many female youth participants overcame diverse personal circumstances to participate in SRH decision-making, such as adolescent pregnancy, gender-based violence and discrimination from their peers, family and communities. This showcased their agency, strength and commitment to be engaged and challenge individual and systemic barriers. Several female youth participants felt comfortable *"voicing out"* on SRH issues and that this reflected progress of changing norms and attitudes. One youth shared how her traumatic experiences motivated her to become a local/global advocate for young women:

> *"I say to myself, 'you know what? I'm the victim of rape or maybe I am a victim of early marriages. . .is it going to predict my future? No!' Being a victim does not mean you can stay idle. Being a victim does not mean you cannot do anything. Being a victim should give you the strength that you're someone strong. You're someone who can do great things. . ..I have to speak out. I have to advocate. I have to make noise." (Female Youth Interview Participant)*

Female youth participants advocated that they should be at the *"forefront"* by participating and meeting their rights and responsibilities. Although they have vital roles in their families and communities, their contributions often lacked recognition. Despite inequities, many female youth participants proclaimed that their gender represents an asset that affords them with unique insights and opportunities to be engaged in SRH decision-making that is unavailable to their male youth counterparts. One female youth participation illustrated the ways that young people have also capitalized on global discourses and momentum around gender equality to create power within SRH decision-making spaces:

> *"Being a female has also helped me to be who I am today. You know? The world is promoting women, so I can say that my gender is also playing a big role in the community—as I am a girl and a youth parliamentarian for that matter." (Female Youth Interview Participant)*

Some male youth also spoke in solidarity with their female peers advocating that young women deserve more equitable representation and recognized that young women's' voices should be more clearly reflected in SRH decision-making processes. A male youth outlined the importance of understanding young women's particular needs and experiences because *"the problem for the males may not be the same for the females."*

The centrality of achieving gender equality to young people's active and meaningful SRH participation, and the urgent need for collective efforts and responsibility to foster equality was visualized by a youth (Fig 3).

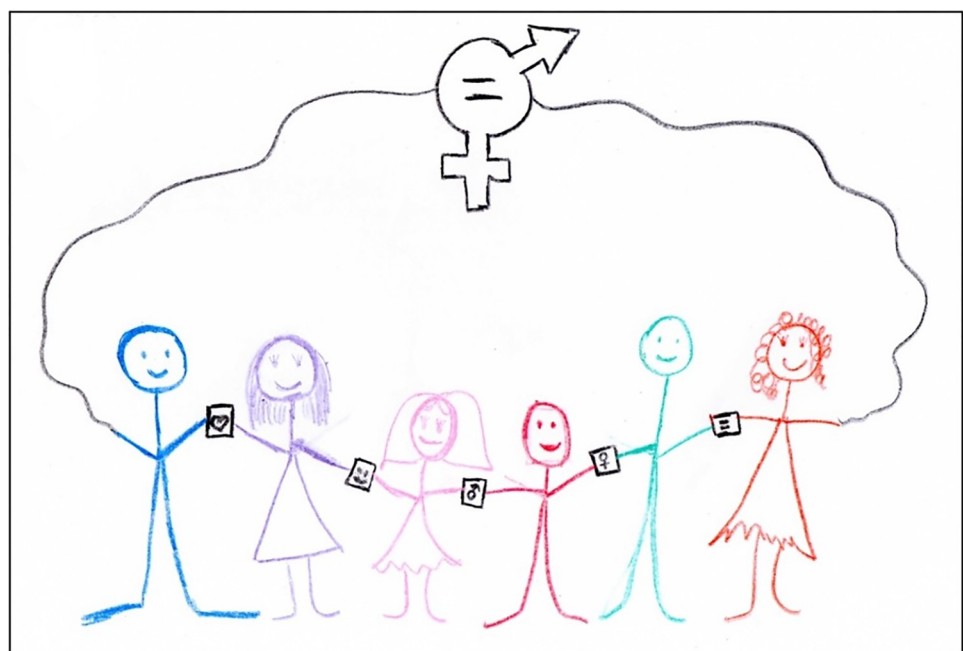

**Fig 3. Achieving gender equality is a collective responsibility.**

## Discussion

### Global rhetoric and local realities

International commitments to youth, SRH and gender equality were articulated through the 2015 adoption of the SDGs, and at a national level through the 2013 introduction of Malawi's *Gender Equality Act*, *National Youth Policy*, and the 2017 constitutional amendments that raised the minimum age of marriage [5, 79, 80]. However, our research reveals a disconnect between the pervasive global and national gender equality rhetoric and young women's lived realities of their SRH and participatory rights in Malawi. Although many participants noted the importance of gender parity within international/national SRH policy spaces (e.g., youth parliament or donor forums), our overall findings indicated that underlying discrimination continues to circumscribe the active and meaningful engagement of young women at all levels of governance, particularly within their own communities. The frequent marginalization of young women's voices in SRH decision-making are shaped by both age and gender discrimination. As such, efforts to achieve gender equity must be grounded in and driven by the lived experiences of young women, in order to achieve significant shifts in their participation.

Participation in both SRH and overall policymaking in Malawi is highly gendered across generations, with underrepresentation of young women in youth participatory spaces and of adult women in democratic governance structures [81, 82]. Women account for just 10% of traditional leaders [82] in Malawi, and as of May 2019, only 44 elected members of Parliament (or 23%) were women [81]. This problem is of course not exclusive to Malawi: inequitable gender representation in National Parliaments remains a global challenge, with most countries (both North and South) failing to realize gender equity in elected bodies. In Canada, women comprise only 29% of elected members of parliament, versus other countries that have achieved gender parity, such as Rwanda (61.3%), Cuba (53.2%) and Bolivia (53.1%) [81]. Research on gender equity and SRH within Malawi's dual (democratic and traditional) governance systems highlights the potential impact of greater women's political engagement, given

that female traditional leaders and members of parliament more effectively promote women's SRH rights than do men [82]. Malawian Senior Chief Theresa Kachindamoto is an example of the importance of female leadership: she has championed both local and national legislation to increase the legal age of marriage and nullified over 1,000 unions that violate these laws [82, 83]. Failure to prioritize young women's participation within key national policies, including the *National Youth Policy*, limit opportunities for capacity building and perpetuates inequities in their engagement moving forward [5].

Our results show that young women's participation and gender equity are interrelated and interdependent. The equal, active, and meaningful SRH participation of young women is contingent on societal recognition and overcoming of gaps in gender equality. Our study extends recent scholarship that has conceptualized the relationship between social norms around gender norms and young people's SRH [84, 85] to also consider the impact of gender inequity on their authentic engagement; whereby young people's voices, and young women's voices in particular, are given equal space and consideration within SRH decision-making processes. As such, SRH policy scholars argue that the continued prioritization of youth is essential to addressing gender (in)equality [86]. This research acknowledges young women's engagement in SRH policymaking, both formal and informal, and represents the critical first step toward overcoming restrictive gender norms.

## Donors and the "girl child"

African critiques around the role and influence of foreign development donors reflects long-standing concerns of neocolonial appropriation of priority-setting and decision-making [87]. In Malawi, international donors contribute over 60% of spending on health care (2012–2015), with funding for SRH and sexually transmitted infections (including HIV/AIDS) representing nearly half (47%) of all external health expenditure [88, 89]. We show that donors' political and ideological priorities profoundly influence the entire SRH policymaking process, which simultaneously proscribes and promotes the engagement of girls and young women. Recently proposed legislation to liberalize abortion (*Termination of Pregnancy Bill*) and the adopted *Adolescent Girls and Young Women Strategy* were identified by informants as examples of "donor-driven" SRH policy efforts.

We expose and acknowledge the patriarchal and colonial legacy of the SRH field, and especially the promotion of contraceptives: funding and implementation of SRH initiatives and policies by former colonial powers is problematic and even suspect as a form of neocolonialism [90]. Participants questioned whether donor-driven SRH policies aligned with Malawian priorities and values, highlighting conflicting neocolonial influence by donors amid local politico-cultural realities. The donor-driven nature of SRH policymaking often means that young women's priorities for SRH information and services are eclipsed by donor ideologies and agendas that introduce global norms within the Malawian context. This illustrates the ways in which the micropolitics of young women's everyday SRH participation are intimately connected to global economic and political processes [17]. Renewed calls to decolonize global health also must work to transform current global rhetoric and approaches within SRH [90, 91].

Notwithstanding these substantial concerns, many youth participants and key informants believed that donors' prioritization of young women has increased their engagement in SRH decision-making processes. This finding contradicts critical literature linking the "turn to the girl" or the "girling of development" to neoliberalism; employing "empowerment" and "agency" discourses are criticized seeking to "rescue" and homogenize young women in the global South [12]. According to critics, girl-oriented development approaches frame girls as

both victims of oppression and heroes with agency [12], whilst placing responsibility on girls to overcome poverty or address SRH issues using only their "voice" or "agency", rather than making structural change through their meaningful participation. Our results also reveal that the donor-driven *"girl child"* discourse in Malawi has triggered young men's hostility and a capitalist, class-based competition towards their female peers, with few men acknowledging the underlying structural forces that often restrict young women's SRH participation. This prioritization of young women, and the neglect of young men was considered ineffective by many, and unintentionally may perpetuate discrimination, undermining youth SRH efforts in Malawi. Continued engagement of both young men and women is needed to address more holistically gender discrimination, norms, and power and health inequities.

## SRH within a "culture" of silence and "stigma"

Study participants associated increased public acceptance of SRH issues with efforts to address high rates of HIV/AIDS in Malawi over the last two decades. Despite perceived progress, SRH and young people's participation in SRH decision-making remains controversial. Older (often male) community members consider young women promiscuous and disrespectful for *"voicing out"*, limiting their ability to speak openly about SRH issues. Moreover, the continued practice of traditional customs preserves patriarchal control over young women's reproduction and bodies, and their economic and decision-making power [92].

   Many youth participants and key informants shared that their religion and culture posed barriers to young women's participation in SRH policymaking. However, postcolonial feminists argue that a focus on "culture" alone is insufficient. Challenging "essentialist interpretations" of the role of culture on young women's lived experiences of participation must consider how it is situated within and shaped by other historical, political, economic and social forces [50]. Our research considers how other underlying influences, including patriarchy, gerontocracy and donors' postcolonial powers, impact the opportunity and quality of young women's SRH participation.

## Patriarchy and postcolonialism

The historical, social and economic contexts of the lives of young women in Malawi have long shaped the ways in which patriarchal gender norms and roles are reproduced. Our study participants explained that men often act as cultural custodians or gatekeepers who control young women's access to SRH information, services, and participation in policymaking spaces. Study participants also pointed to the tokenistic or symbolic engagement of young women in community youth clubs and district decision-making structures. This suggests that a large proportion of young women may be privy to SRH decision-making as permitted by male custodians. Recent empirical evidence from other previously colonized countries (e.g., Senegal and Kenya) also found that formal/informal opportunities for youth to participate meaningfully, often reproduced underlying discourses of gender (in)equality and patriarchy, gendered norms and constructions of citizenship [93, 94].

## Young women as "second-class" citizens

Young people's intersectional social identities shape their lived experiences of participation. Older youth, males, young people living in urban areas, and those with more formal education have greater opportunities than younger, female, less educated and rural residents to participate in SRH policymaking in Malawi [6]. Although all but two of the youth participants in our study were 18 years of age or older, many shared significant experiences of age-based discrimination and marginalization prior to and during their involvement in SRH decision-making

processes. Literature on engagement of young women in the United Nations Commission on the Status of Women also highlights that the intersection of age and gender reinforced their political marginalization [10]. Overlapping and systemic patriarchal and adultist norms highlight the ambiguity of the temporality of social identities and young women's transition into adulthood; several participants associated transitioning into adulthood with young women's marital and childbearing status. Young women are often relegated to "second-class" citizen status due to their systemic discrimination and failure to protect and realize their basic participatory and SRH rights [95]. These shortcomings highlight the importance of challenging normative perceptions of young women's status roles in society, by generating contextualized insights of young people's lived experiences of SRH participation. Efforts are needed to ensure the continued engagement of young women who are married or have children; their perspectives and priorities are underrepresented in current SRH decision-making processes. Further evidence is also needed to reflect younger adolescent girls' experiences of SRH participation, as they are likely to encounter even greater hurdles to engage actively and meaningfully.

Youth engagement in SRH decision-making and gender equality represented shared priorities among both male and female youth participants. Participants emphasized a consensus that young women experienced inequitable barriers, underscoring that age, gender, race, and class interact to challenge their participation [54]. Both groups reported that young men's perspectives were preferred in SRH policymaking, and that judgment/shame associated with participating in decision-making or accessing SRH services was reserved for young women. Both male and female youth reported that young women were uncomfortable speaking about SRH in policy spaces, particularly with older, male decision-makers.

Our findings underscore the substantial challenges to effectively democratize SRH decision-making power to poor and female Malawian youth. Multiple participants emphasized the inequitable power and socioeconomic status of young women, which threatened their capacity to participate and realize their SRH and participatory rights. Inadequate knowledge of policymaking or ability to read and write English represented critical barriers to their engagement. Creating open and authentic policy spaces, and training and mobilizing all youth to challenge this "hypocrisy" and cycle of patriarchal and gerontocratic systems is critical to addressing the inequitable participation and SRH among young women [96].

## Young women as agents of SRH change

Opportunities for young women to participate and lead in SRH decision-making demonstrate both progress and potential for their engagement in Malawi. Despite encountering structural forces, young women emerged in our study as passionate community role models and agents of SRH change, motivated by their duty to support their community and country [6]. Several young female participants overcame personal and systemic barriers to pursue educational opportunities and SRH participation. Our PCF/DCA perspective emphasizes that recognition of young women's heterogeneous experiences are critical to disrupting the systems of power that marginalize their SRH engagement [50, 97]. Through both formal (e.g., national youth parliament) and informal participation (e.g., advocacy to increase the age of marriage), young women in our study demonstrated significant capacity, knowledge of SRH issues, and agency to lay the foundation for a grassroots movement to ensure their active and meaningful involvement in SRH decision-making.

Our findings illustrate that young people are shaped by their economic, cultural, and social contexts, yet simultaneously resist dichotomies of power [29]. To many of our female youth participants, being a young woman represented an asset to their participation. This perspective confronts normative understandings of citizenship by advocating for the recognition and

celebration of "difference" [54]. This was illustrated by a female youth as she resisted patriarchal attitudes and power inequities *"when people see me, they think that being a woman doesn't mean that we are inferior, that we can also bring change to the world."* The recent global/national prioritization of young women and youth participation has also created promising opportunities for their engagement in decision-making, which must continue to be leveraged to mobilize sustainable changes to existing gendered and adultist norms.

## Research implications

Our study finds that underlying gendered structural and systemic forces continue to challenge progress towards achieving global goals pertaining to young women's SRH and their meaningful participation in SRH policymaking (and beyond). Many participants were interested in how youth in other countries (e.g., Canada) are engaged in SRH policymaking and these discussions highlighted the comparative progress and promising policy efforts for youth participation in Malawi, versus some countries in the global North [98]. As a result, we argue that engaging young people in SRH decision-making, particularly young women, represents a challenge that is not unique to Malawi, but a broader global health and human rights concern.

Recent global developments, including the COVID-19 pandemic, reinforce the urgency of addressing young people's SRH and ensuring their inclusion in decision-making processes. Decades of progress and efforts targeting gender equality and SRH are at risk, as many countries in the global South, such as Malawi, struggle to ensure accessibility of contraception, skilled health providers, and YFHS amidst competing demands on the health system [99]. School closures and public health restrictions in many sub-Saharan African countries, including Malawi, already have been linked to increases in adolescent pregnancy [100]. A rapid assessment conducted by the government highlighted that teenage pregnancies increased by 11% from March to July 2020, as compared to the same time period in 2019. Key drivers identified were religious beliefs, inadequate economic and social opportunities and a desire to start childbearing [101]. In addition, research examining school enrollment pre- and post-school closures found that the highest rate of dropouts were amongst older female youths, aged 17–19 years, exacerbating existing gender inequalities in education [102]. This suggests that COVID-19 represents not only a current health threat, but is likely to have long-lasting impact on young people's SRH and participation. Our work reinforces that addressing the systemic silencing of young women is crucial to ensure that SRH policymaking processes reflect young people's lived realities, now more than ever.

## Study limitations

Despite efforts to ensure representation of diverse youth perspectives, few young adolescents (under 18 years), young people with disabilities, or lesbian, gay, transgender, bisexual or queer (LGTBQ) youth were recruited. In addition, ethnicity and cultural or tribal membership was not explicitly considered during sampling of research participants. This represents a potential limitation, particularly as some cultural practices are associated with specific tribal or religious groups. We anticipated that the geographic range of sampling would capture diverse perspectives, contexts, traditions and practices within the country. Further analysis is needed of the influence of ethnicity or tribal affiliations on the power and status of young women, and their involvement in SRH decision-making.

In addition, although we aimed to be inclusive and recruit diverse youth perspectives, we acknowledge that this research and focus on youth SRH in Malawi was influenced by underlying heteronormative assumptions. In Malawi, a country that has not legalized same-sex relations, many LGBTQ communities face discrimination, violence and the threat of criminal

penalty for failing to conform to heteronormative standards [103]. Adopting a postcolonial queer feminist perspective may be necessary to challenge heteronormative structures, which limit the participation of individuals who don't fall within binary categories of sex/gender [104]. Specifically, research that is inclusive of those with diverse gender and sexual identities and orientations, yet culturally grounded (and legal), is needed to understand the SRH needs of these groups and the role of heteronormative assumptions in shaping youth participation in SRH decision-making processes in Malawi and beyond.

## Conclusion

Significant global development discourse and national policy efforts have focused on gender equality and youth participation in SRH. However, the lived experiences of young women participating in SRH policymaking in Malawi demonstrate entrenched power structures that privilege men, adults, and individuals in higher social economic status. Understanding how gender and age together shape young people's involvement in Malawi will help inform contextualized and tailored strategies to support the participation of young women in SRH decision-making. Young women experience gendered participation in SRH policymaking at all levels, as they are too often underrepresented and undervalued. Despite these systemic barriers young women also represent critical resources and local agents of change for youth SRH. Young women's vital roles and contributions in SRH policymaking must be recognized through continued advocacy and education of their peers, families, communities and decision-makers. Our use of postcolonial feminism aims to inspire "solidarity across borders" [17]. The lessons from progress and challenges to engaging young women in SRH policymaking in Malawi are applicable across both the global South and North. Gender equality and youth participation represent distinct areas within global and national development processes, yet our research shows that they are clearly interdependent. The active and meaningful participation of young women in Malawi (and beyond) is not possible without collective efforts to catalyze political, societal, and cultural change to challenge systemic gender and power inequities.

## Supporting information

**S1 File. Inclusivity questionnaire.**
(DOCX)

## Acknowledgments

Thank you to the youth participants who generously shared their time, energy and experiences during this study, without you this research would not have been possible. We greatly appreciate the engagement of youth researchers: Brenda Mweso, Dan Chakanika-Phiri, Eric Banda, Innocent Mkangama and Memory Nyimbiri. In addition, thank you to the key informants who contributed important insights and perspectives to this work. We acknowledge the support from our local host institution, PACHI, and Charles Makwenda.

This article is dedicated to the memory of our co-author Paula Braitstein, whose commitment as a global health scholar and advocate inspired us all.

## Author Contributions

**Conceptualization:** Jannah M. Wigle.

**Data curation:** Jannah M. Wigle, Stewart Paul, Monica Kalolo.

**Formal analysis:** Jannah M. Wigle.

**Funding acquisition:** Jannah M. Wigle.

**Investigation:** Jannah M. Wigle, Stewart Paul.

**Methodology:** Jannah M. Wigle, Anne-Emanuelle Birn, Brenda Gladstone.

**Project administration:** Jannah M. Wigle, Stewart Paul, Lumbani Banda.

**Software:** Jannah M. Wigle.

**Supervision:** Anne-Emanuelle Birn, Brenda Gladstone, Paula Braitstein.

**Writing – original draft:** Jannah M. Wigle.

**Writing – review & editing:** Jannah M. Wigle, Stewart Paul, Anne-Emanuelle Birn, Brenda Gladstone, Monica Kalolo, Lumbani Banda, Paula Braitstein.

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
