## [Decision Letter · Decision Letter 0]

8 Aug 2022

PGPH-D-22-00499

Participation of young women in sexual and reproductive health decision-making in Malawi: Local realities versus global rhetoric

Dear Dr. Wigle,

Thank you for submitting your manuscript to PLOS Global Public Health. After careful consideration, we feel that it has merit but does not fully meet PLOS Global Public Health’s publication criteria as it currently stands. Therefore, we invite you to submit a revised version of the manuscript that addresses the points raised during the review process.

We look forward to receiving your revised manuscript.

Kind regards,

Melanie Boeckmann

Academic Editor

Journal Requirements:

3. Please provide separate figure files in .tif or .eps format and removed from the manuscript file.

4. Figs 1-3: Please confirm (a) that you are the photographer; or (b) provide written permission from the photographer to publish the photo(s) under our CC-BY 4.0 license.

4. In the online submission form you indicate that your data is not available for proprietary reasons and have provided a contact point for accessing this data. Please note that your current contact point is a co-author on this manuscript. According to our Data Policy, the contact point must not be an author on the manuscript and must be a third party. Please revise your data statement to a non-author institutional point of contact, such as a data access or ethics committee, and send this to us via return email. Please also include contact information for the third party organization, and please include the full citation of where the data can be found.

Additional Editor Comments (if provided):

Reviewers' comments:

Reviewer's Responses to Questions

**Comments to the Author**

1. Does this manuscript meet PLOS Global Public Health’s publication criteria? Is the manuscript technically sound, and do the data support the conclusions? The manuscript must describe methodologically and ethically rigorous research with conclusions that are appropriately drawn based on the data presented.

Reviewer #1: Yes

Reviewer #2: Yes

2. Has the statistical analysis been performed appropriately and rigorously?

Reviewer #1: N/A

Reviewer #2: N/A

3. Have the authors made all data underlying the findings in their manuscript fully available (please refer to the Data Availability Statement at the start of the manuscript PDF file)?

Reviewer #1: No

Reviewer #2: Yes

4. Is the manuscript presented in an intelligible fashion and written in standard English?

Reviewer #1: Yes

Reviewer #2: Yes

5. Review Comments to the Author

Reviewer #1: Thank you for the chance to review this manuscript. It is interesting work and the commitment to open, reflexive, and feminist research is great to see. Such research will be beneficial to policymakers and programmers. I attach comments below and eagerly await the thoughts from the authors.

The title of this is about SRH decision-making, but sometimes it is not always clear how you are defining participation and decision-making (which can have an enormous array of meanings in SRHR research).

- Line 37 – make explicit that this is not about (as far as I understand it) participation in daily SRH decisions? But is about SRH at the political level.

It could be worth acknowledging that the category of youth, as defined in Malawian policy, also allows for flexibility in age markers (https://www.unicef.org/esa/media/9031/file/UNICEF-Malawi-2020-2021-Youth-Budget-Brief.pdf)

- In a later point (line 493) you talk about when a young woman is considered a youth, and how this intersects with her sexual and reproductive lifecourse. Grappling with this could be brought out further, particularly the incongruity of what the government determine to be a youth compared to the contextual realities (in which markers other than age are involved).

Recommend reading African Sexualities: A Reader (ed. Sylvia Tamale) – has interesting and important work on gender and sexuality across Africa

210: six youth researchers engaged – how? What was this process? Could there be any reflection on the potential age differences between youth researchers and participants?

257-58: What is meant by generating data “from youth and key informants’’ perspectives”?

Results

Gender Equality Rhetoric: could this section be linked more specifically to SRH? There is the quote about chiefs saying that girls cannot be leaders – what is this in reference to? Does participation in youth parliament, for example, matter when it comes to SRH? It definitely has an important implication for general political participation, but could be useful to be clear how this relates to your RQs.

Under Donor Influence on Youth and SRH Policy: Why does it matter that there are anti-abortion groups in relation to the Termination of Pregnancy Bill? How does this relate to youth participation in SRH decision-making?

530-532: “Many female youth participants overcame barriers to participate in SRH decision-making, such as adolescent pregnancy, gender-based violence and discrimination from their peers, family and communities.”

- What are these barriers? Adolescent pregnancy is represented as a barrier, what if an adolescent is pregnant people they had full SRH decision-making? There’s a value judgement here that needs explaining.

Quote 548-550 – “the world is promoting women” – ties to earlier points about the global discourse? Does this show the utilisation by young people of global discourses to create power within their community spaces?

Discussion

608-609: “We show that donors’ political and ideological priorities profoundly influence the entire SRH policymaking process” – repeated on 613-614

646: “Efforts to address high rates of HIV/AIDS and maternal mortality in Malawi…were associated with an increased public acceptance of SRH issues by study participants” – I don’t think this evidence has been shown? Perhaps speaks to the point below about your methods.

748 = “drastic” increases in adolescent pregnancy is potentially stigmatising language. Would be better to show evidence that these pregnancies were not wanted or a choice among adolescents

In your limitations – you say that this research is rooted in heteronormative assumptions – and then define terms. Can you be explicit in how your research was rooted in these assumptions and why? Was anything done to counteract the impact of this?

- I think it would be important to grapple with some of these ideas, considering the intention to “map the margins” in this research. Using Collins and Crenshaw, what are the limits of your ability to do this? Tying back to this would complete the narrative and show engagement with your work

Methods

Some of the methods that you use feel underdeveloped in the paper.

The document analysis of policies in particular feels underused, and it reduces the impact of the novelty of your work. Taking your theoretical framework and use of an intersectional lens in your analysis, what do the documents tell us specifically about e.g., how ‘youths’ are defined, how do policies talk about women, who is included / excluded / not mentioned? Extending your theoretical approach in your document analysis would bolster the discussion, which at times does not feel as novel as it could.

The data from focus groups is often presented similarly to data gathered from in-depth interviews. This doesn’t maximise the uniqueness of focus group discussions at capturing how concepts and topics are discussed between participants, and the insights we can gain from these interactions.

Reviewer #2: This is a timely and important contribution to the literature around gendered and generational participation in policy decision-making in Malawi. The authors argue that young women are shut out of policy-making through a confluence of intersecting factors including culturally structured and approved heterosexism and ageism (in the form of a gerontocracy). Drawing on feminist and decolonial theories of intersectionality and difference, the authors point out the dynamic nature of Malawian society, especially among the youth, to illustrate the multiple ways in which young people resist and remake these cultural norms to promote young women’s participation in health governance both formally and informally. As such, it contributes to the literature on youth agency, Southern African societies, and sexual and reproductive health policy-making dynamics.

I have approached the article with an attitude of curiosity and good faith. I have offered extensive comments below because I believe in this author’s capability and the importance of the topic under examination. My main suggestion to the author is to draw out explicitly the creation and direction of power differentials in order to highlight how theory is played out in practice. As a medical anthropologist working in global health, I also want to see us collectively move beyond jargon in order to press the utility of these theories further. (This means owning one’s voice in new ways.) Doing so will move the article to another level. I note where this can be done in the appropriate lines below. The author has prompted me to think, and I want to return the favour.

To move in this direction, I therefore ask for a gloss on several terms to point in that direction. For example, defining inequity (vs inequality) in the Table of Concepts would also help orient the reader to the author’s use of this often ambiguous term. I suggest placing this after ‘Gender’ and before ‘Gender Equity’ in the table. (I particularly liked the inclusion of a Table of Key Concepts.)

In a more minor vein, the article would benefit from some tidying up. Style-wise, the author’s extensive use of “/” between words that are sometimes synonymous and sometimes rather distinct dilutes the author’s argument and reduces clarity for the reader. I’ve tried to suggest alternatives where appropriate. Though word count is not an issue for this journal, I have also tried to make those suggestions in ways that promote conciseness.

Similarly, the author has used passive voice in several areas which diminishes the agency of the participants. I encourage active voice. In the press, use of passive voice is a tool of institutional oppression designed to exculpate institutional actors from any personalised, localized agency because it hides from view the actual subject creating or reinforcing oppressive structures. The passive voice can also deny visible agency to those of lower status for a similar reason: it obscures their own decision to act. Because of the author’s stated theoretical stance in decolonial feminist approaches to global health, I suggest the authors avoid use of the passive voice whenever possible.

In the attachment, I listed my suggestions in line order, but have also indicated whether they concern ‘style’ (which includes any typos or grammatical oversights), ‘theory’, and ‘other’ in the lead of each suggestion. Theory and other are major points I believe the author should address before publication; style is minor, and simply a result of my close reading of the manuscript to improve clarity.

Overall, I found the article interesting and insightful. I look forward to seeing it published.

6. PLOS authors have the option to publish the peer review history of their article (what does this mean?). If published, this will include your full peer review and any attached files.

**Do you want your identity to be public for this peer review?** For information about this choice, including consent withdrawal, please see our Privacy Policy.

Reviewer #1: No

Reviewer #2: **Yes: **Jason Johnson-Peretz

---

## [Decision Letter · Decision Letter 1]

26 Oct 2022

Participation of young women in sexual and reproductive health decision-making in Malawi: Local realities versus global rhetoric

PGPH-D-22-00499R1

Dear Dr Wigle,

We are pleased to inform you that your manuscript 'Participation of young women in sexual and reproductive health decision-making in Malawi: Local realities versus global rhetoric' has been provisionally accepted for publication in PLOS Global Public Health.

Best regards,

Melanie Boeckmann

Academic Editor

Reviewer Comments (if any, and for reference):

Reviewer's Responses to Questions

**Comments to the Author**

1. If the authors have adequately addressed your comments raised in a previous round of review and you feel that this manuscript is now acceptable for publication, you may indicate that here to bypass the “Comments to the Author” section, enter your conflict of interest statement in the “Confidential to Editor” section, and submit your "Accept" recommendation.

Reviewer #1: All comments have been addressed

Reviewer #2: All comments have been addressed

2. Does this manuscript meet PLOS Global Public Health’s publication criteria? Is the manuscript technically sound, and do the data support the conclusions? The manuscript must describe methodologically and ethically rigorous research with conclusions that are appropriately drawn based on the data presented.

Reviewer #1: Yes

Reviewer #2: Yes

3. Has the statistical analysis been performed appropriately and rigorously?

Reviewer #1: N/A

Reviewer #2: N/A

4. Have the authors made all data underlying the findings in their manuscript fully available (please refer to the Data Availability Statement at the start of the manuscript PDF file)?

Reviewer #1: (No Response)

Reviewer #2: Yes

5. Is the manuscript presented in an intelligible fashion and written in standard English?

Reviewer #1: Yes

Reviewer #2: Yes

6. Review Comments to the Author

Reviewer #1: Thank you for addressing the points raised. I am looking forward to this being published

Reviewer #2: Great job!

7. PLOS authors have the option to publish the peer review history of their article (what does this mean?). If published, this will include your full peer review and any attached files.

**Do you want your identity to be public for this peer review?** For information about this choice, including consent withdrawal, please see our Privacy Policy.

Reviewer #1: No

Reviewer #2: **Yes: **Jason Johnson-Peretz
